# MULTI-LEVEL RESIDUAL NETWORKS FROM DYNAMICAL SYSTEMS VIEW

**Bo Chang**[*]**, Lili Meng**[*] **& Eldad Haber**
University of British Columbia & Xtract Technologies Inc.
Vancouver, Canada
{bchang@stat, menglili@cs, haber@math}.ubc.ca

**Frederick Tung**
Simon Fraser University
Burnaby, Canada
ftung@sfu.ca

**David Begert**
Xtract Technologies Inc.
Vancouver, Canada
david@xtract.ai

## ABSTRACT

Deep residual networks (ResNets) and their variants are widely used in many computer vision applications and natural language processing tasks. However, the theoretical principles for designing and training ResNets are still not fully understood. Recently, several points of view have emerged to try to interpret ResNet theoretically, such as unraveled view, unrolled iterative estimation and dynamical systems view. In this paper, we adopt the dynamical systems point of view, and analyze the lesioning properties of ResNet both theoretically and experimentally. Based on these analyses, we additionally propose a novel method for accelerating ResNet training. We apply the proposed method to train ResNets and Wide ResNets for three image classification benchmarks, reducing training time by more than 40% with superior or on-par accuracy.

## 1 INTRODUCTION

Deep neural networks have powered many research areas from computer vision (He et al., 2016; Huang et al., 2017b), natural language processing (Cho et al., 2014) to biology (Esteva et al., 2017) and e-commerce (Ha et al., 2016). Deep Residual Networks (ResNets) (He et al., 2016), and their variants such as Wide ResNets (Zagoruyko & Komodakis, 2016) and DenseNets (Huang et al., 2017b), are among the most successful architectures. In ResNets, the authors employ *identity skip-connections* that bypass residual layers, allowing data to flow from previous layers directly to any subsequent layers.

With the success of ResNet and its variants on various applications (He et al., 2016; 2017; Pohlen et al., 2017; Xiong et al., 2017; Oord et al., 2016; Wu et al., 2016), several views such as *unraveled view* (Veit et al., 2016), *unrolled iterative estimation view* (Greff et al., 2017) and *dynamical systems view* (Haber et al., 2017; E, 2017; Chang et al., 2017) have emerged to try to interpret ResNets through theoretical analysis and empirical results. These views provide preliminary interpretations, however, deep understanding of ResNets is still an active on-going research topic (Jastrzebski et al., 2017; Li et al., 2016; Hardt & Ma, 2017; Li & Yuan, 2017).

The dynamical systems view interprets ResNets as ordinary differential equations (ODEs), a special kind of dynamical systems (Haber et al., 2017; E, 2017), opening up possibilities of exploiting the computational and theoretical success from dynamical systems to ResNets. From this point of view, stable and reversible architectures (Haber & Ruthotto, 2017; Chang et al., 2017) are developed. However, few empirical analysis of this view has been done and many phenomena such as the removing of layers not leading to performance drop are not explained by the dynamical systems view. In this work, we take steps forward to complement this dynamical systems view with empirical analysis of its properties and the lesion studies.

---

[*]Authors contributed equally.

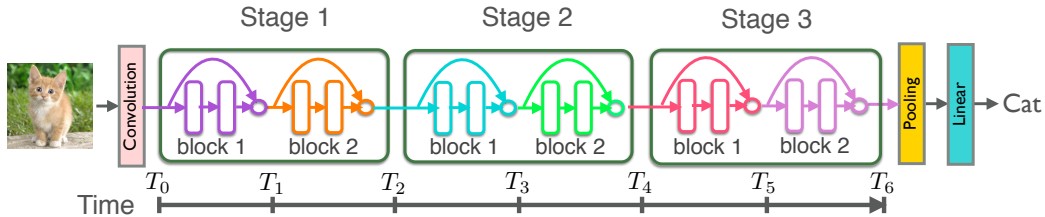

**Figure 1: Dynamical systems view of ResNets.** ResNets equally discretize the time interval $[0, T]$ using time points $T_0, T_1, \ldots, T_d$, where $T_0 = 0$, $T_d = T$ and $d$ is the total number of blocks.

One challenge of deep ResNets is the long training time. It is extremely time-consuming to train on large datasets such as ImageNet or with very deep ResNets such as 1000-layer networks, which may take several days or even weeks on high-performance hardware with GPU acceleration. Recently the reversible residual networks (Gomez et al., 2017; Chang et al., 2017) consumes 50% more computational time for reducing memory usage by reconstructing the activations, which exposes training time to be a more severe problem. Inspired by the dynamical systems interpretation, we additionally propose a simple yet effective *multi-level* method for accelerating ResNets training.

In summary, the main contributions of this work are:

- We study the dynamical systems view and explain the lesion studies from this view through empirical analysis.

- We propose a simple yet efficient multi-level training method for ResNets based on dynamical systems view.

- We demonstrate the proposed multi-level training method on ResNets (He et al., 2016) and Wide ResNets (Zagoruyko & Komodakis, 2016) across three widely used datasets, achieving more than 40% training time reduction with superior or on-par accuracy.

## 2 RELATED WORK

### 2.1 RESNETS AND VARIANTS

ResNets (He et al., 2016) are deep neural networks of stacking simple residual blocks, which contain identity skip-connections that bypass the residual layers. A **residual block**, as shown in Fig. 2, can be written as

$$\mathbf{Y}_{j+1} = \mathbf{Y}_j + G(\mathbf{Y}_j, \theta_j) \quad \text{for} \quad j = 0, \ldots, N-1, \tag{1}$$

where $\mathbf{Y}_j$ is the feature map at the $j$th layer, $\theta_j$ represents the $j$th layer's network parameters. $G$ is referred to as a **residual module**, consisting of two convolutional layers. As shown in Fig. 1, the network is divided into several stages; each consists of a number of residual blocks. In the first block of each stage, the feature map size is halved, and the number of filters is doubled. The feature map remains the same dimensionality for subsequent blocks in a stage.

**Figure 2: A residual block.** Each residual block has two components: the residual module $G$ and the identity skip-connection, which add up to the output of a block.

After the success of ResNets in popular competitions such as ImageNet (Russakovsky et al., 2015), Pascal VOC (Everingham et al., 2010) and Microsoft COCO (Lin et al., 2014), there emerged many successors (Huang et al., 2016; 2017b; Chang et al., 2017; Gomez et al., 2017; Targ et al., 2016; Hardt & Ma, 2017; Zagoruyko & Komodakis, 2016). For instance, DenseNet (Huang et al., 2017b) connects between any two layers with the same feature-map size. ResNxt (Xie et al., 2017) introduces a homogeneous, multi-branch architecture to increase the accuracy.

## 2.2 INTERPRETATIONS OF RESNETS

**Unraveled view** In Veit et al. (2016), ResNets are interpreted from an unraveled view in which ResNets are viewed as a collection of many paths which data flow along from input to output. Each residual block consists of a residual module and an identity skip-connection; a path is defined as a configuration of which residual module to enter and which to skip. For a ResNet with $n$ residual blocks, there are $2^n$ unique paths. Through lesion studies, Veit et al. (2016) further demonstrate that paths in ResNets do not strongly depend on each other and behave like an ensemble. When a residual block is removed, the number of paths is reduced from $2^n$ to $2^{n-1}$, leaving half of the paths still valid, which explains why ResNets are resilient to dropping blocks. Besides the explicit ensemble view, training ResNets with stochastic depth (Huang et al., 2016) can be viewed as an ensemble of networks with varying depths implicitly.

**Unrolled iterative estimation view** ResNets are interpreted as unrolled iterative estimation in Greff et al. (2017). From this view, the level of representation stays the same within each stage. The residual blocks in a stage work together to estimate and iteratively refine a single level of representation: the first layer in a stage provides a rough estimate for the representation, and subsequent layers refine that estimate. An implication of this view is that processing in each block is incremental and removing blocks only has a mild effect on the final results. Based on this view, Jastrzebski et al. (2017) provide more analytical and empirical results.

**Dynamical systems view** ResNets can be interpreted as a discretization of dynamical systems (Haber et al., 2017; E, 2017). The basic dynamics at each step is a linear transformation followed by component-wise nonlinear activation function. The behavior of large dynamical systems is often a notoriously difficult problem in mathematics, particularly for discrete dynamical systems. This is similar to the gradient exploding/vanishing problem for deep neural networks or recurrent neural networks. Imposing structural constraints on dynamical systems such as Hamiltonian systems to conserve the energy is explored in Haber & Ruthotto (2017); Chang et al. (2017). However, no interpretation on the phenomenon of deleting layers is studied from this point of view.

## 2.3 RESNETS EFFICIENT TRAINING METHODS

One major challenge of deep ResNets is their long training time. To alleviate this issue, several attempts have been made. Stochastic depth (Huang et al., 2016) randomly drops entire residual blocks during training and bypassing their transformations through identity skip-connections; during testing, all the blocks are in use. When a block is bypassed for a specific iteration, there is no need to perform forward-backward computation. With stochastic depth, approximately 25% of training time could be saved. Figurnov et al. (2017) reduce the inference time of residual networks by learning to predict early halting scores based on the image content. Huang et al. (2017a) investigate image classification with computational resource limits at test time. SparseNets (Zhu et al., 2018) is a simple feature aggregation structure with shortcut paths bypassing exponentially growing number of layers. Mollifying networks (Gulcehre et al., 2016) start the optimization with an easier (possibly convex) objective function and let it evolve during the training, until it eventually goes back to being the original, difficult to optimize, objective function. It can also be interpreted as a form adaptive noise injection that only depends on a single hyperparameter.

## 3 RESNETS FROM DYNAMICAL SYSTEMS VIEW

In this section, we first provide a brief introduction to the dynamical systems view in which ResNets are considered as ODEs. Based on this view, we provide empirical analysis to explain some intriguing properties and phenomena of ResNets.

### 3.1 DYNAMICAL SYSTEMS VIEW

For the pre-activation ResNets $\mathbf{Y}_{j+1} = \mathbf{Y}_j + G(\mathbf{Y}_j, \boldsymbol{\theta}_j)$, the residual module $G$ consists of two sets of batch normalization, ReLU and convolutional layers. Without loss of generality, we can conceptually add a parameter $h$ and rewrite the residual module as $G = hF$. The residual block becomes

$$\mathbf{Y}_{j+1} = \mathbf{Y}_j + hF(\mathbf{Y}_j, \boldsymbol{\theta}_j), \tag{2}$$

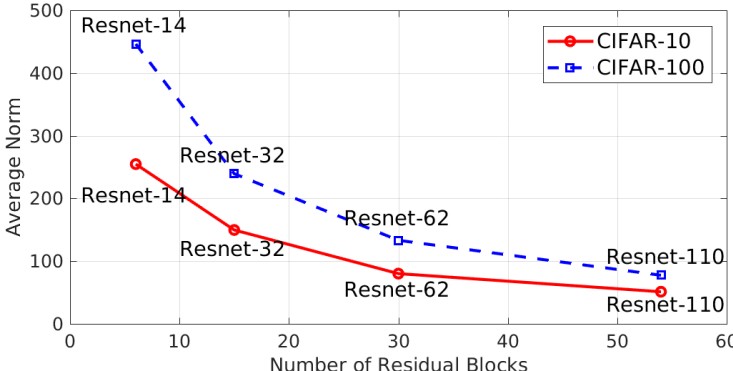

Figure 3: **The average $L^2$-norm of the residual modules $\gamma$ vs the number of residual blocks $d$.** The curve resembles a reciprocal function, which is consistent with Eq. (5) and the dynamical systems view.

which can be further rewritten as

$$\frac{\mathbf{Y}_{j+1} - \mathbf{Y}_j}{h} = F(\mathbf{Y}_j, \theta_j). \tag{3}$$

For a sufficiently small $h$, Eq. (3) can be regarded as a forward Euler discretization of the initial value ODE

$$\dot{\mathbf{Y}}(t) = F(\mathbf{Y}(t), \theta(t)), \ \mathbf{Y}(0) = \mathbf{Y}_0, \text{ for } 0 \leq t \leq T, \tag{4}$$

where time $t$ corresponds to the direction from input to output, $\mathbf{Y}(0)$ is the input feature map after the initial convolution, and $\mathbf{Y}(T)$ is the output feature map before the softmax classifier. Thus, the problem of learning the network parameters, $\theta$, is equivalent to solving a parameter estimation problem or optimal control problem involving the ODE in Eq. (4).

## 3.2 TIME STEP SIZE

The new parameter $h$ is called the **step size** of discretization. In the original formulation of ResNets in Eq. (1), $h$ does not exist, and is implicitly absorbed by the residual module $G$. We call it the **implicit step size**. The step size $h$ can also be explicitly expressed in the model: the output of the residual module is multiplied by $h$ before being added to the identity mapping. In this case, $h$ is a hyper-parameter and we name it the **explicit step size**. In this section, we only consider implicit step size.

We assume that $\mathbf{Y}(0)$ and $\mathbf{Y}(T)$ correspond to the input and output feature maps of the network respectively, where the time length $T$ is fixed. As illustrated in Fig. 1, ResNets equally discretize $[0, T]$ using time points $T_0, T_1, \ldots, T_j, \ldots, T_d$, where $T_0 = 0$, $T_d = T$ and $d$ is the number of blocks. Thus each time step is $h = T_{j+1} - T_j = T/d$.

Thus, we can obtain

$$\|G(\mathbf{Y}_j)\| = \|hF(\mathbf{Y}_j)\| = \frac{T}{d}\|F(\mathbf{Y}_j)\|, \tag{5}$$

where $F$ in the underlying ODE in Eq. (4) does not depend on $d$. In other words, *d is inversely proportional to the norm of the residual modules $G(Y_j)$.*

**Empirical analysis** To verify the above statement, we run experiments on ResNets with varying depths. If our theory is correct, the norm of the residual module $\|G(\mathbf{Y}_j)\|$ should be inversely proportional to the number of residual blocks. Take ResNet-32 with 15 residual blocks in total as an example. We calculate the average $L^2$-norm of the residual modules $\gamma = \frac{1}{15}\sum_{j=1}^{15}\|G(\mathbf{Y}_j)\|$. Figure 3 shows $\gamma$ for different ResNet models. The curve resembles a reciprocal function, which is consistent with Eq. (5) and the dynamical system point of view.

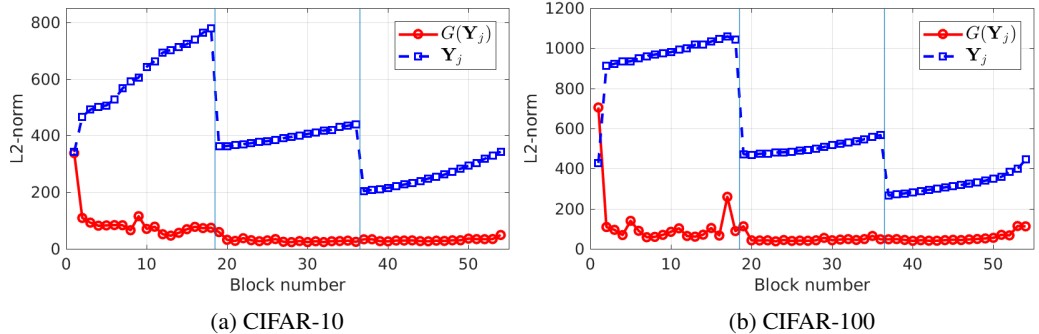

(a) CIFAR-10           (b) CIFAR-100

Figure 4: $L^2$-**norm of the input and output of the residual module** $G$. ResNet-110 models are trained on CIFAR-10 and CIFAR-100. The norms are evaluated at test time. It shows that within a residual block, the identity mapping contributes much more than the residual module. In other words, $G(\mathbf{Y}_j)$ is relatively small for most residual blocks.

### 3.3 LESION STUDY

Lesion studies for ResNets in Veit et al. (2016) remove single or multiple residual blocks, and shuffle the residual blocks at test time. Surprisingly, only removing downsampling blocks has a modest impact on performance, no other block removal leads to a noticeable effect.

According to the dynamical systems view, removing one residual block is equivalent to skipping one time step and squeezing two adjacent steps into one. This operation may change the dynamical system. However, we will show in the following that *the effect is negligible when the output of residual module* $G(\mathbf{Y}_j)$ *is small enough.*

Let $t_0, t_1, t_2$ be three consecutive time points such that $t_1 = t_0 + h$ and $t_2 = t_0 + 2h$. Suppose the removed block corresponds to time point $t_1$. Before the removal of the time point, the discretization is

$$\mathbf{Y}(t_1) = \mathbf{Y}(t_0) + hF(\mathbf{Y}(t_0)),$$
$$\mathbf{Y}(t_2) = \mathbf{Y}(t_1) + hF(\mathbf{Y}(t_1)) = \mathbf{Y}(t_0) + hF(\mathbf{Y}(t_0)) + hF(\mathbf{Y}(t_1)). \tag{6}$$

After $t_1$ is removed, the time interval $[t_0, t_2]$ is squeezed to $[t_0, t_2']$, where $t_2' = t_0 + h$ is the new time point after $t_0$. The new discretization is

$$\mathbf{Y}(t_2') = \mathbf{Y}(t_0) + hF(\mathbf{Y}(t_0)). \tag{7}$$

The difference of the feature before and after the removal operation is

$$\mathbf{Y}(t_2') - \mathbf{Y}(t_2) = hF(\mathbf{Y}(t_1)), \tag{8}$$

which is the output of the residual module $G(\mathbf{Y}(t_1))$. Therefore, the effect of removing the block is negligible when $G(\mathbf{Y}(t_1))$ is small.

**Empirical analysis** To empirically verify that $G(\mathbf{Y}(t))$ are small, we train a ResNet-110 model (3 stages with 18 residual blocks per stage) on CIFAR-10/100, and plot the $L^2$-norm of input $\mathbf{Y}_j$ and output $G(\mathbf{Y}_j)$ of each residual module at test time. As shown in Figure 4, except for the first block at each stage, later blocks have tiny residual module outputs $G(\mathbf{Y}_j)$ compared with the inputs $\mathbf{Y}_j$. This provides an explanation why removing one block does not notably impact the performance.

The effect of shuffling residual blocks can be analyzed in a similar way. When the outputs of the residual modules are small, each block only slightly modifies the feature map. Therefore, we can expect the effect of shuffling to be moderate, especially in later stages.

When the network is deep, the outputs of the residual modules are close to zero. Each residual module can be regarded as feature refinement. The magnitude of change is large only in the first block; the subsequent blocks only slightly refine the features, which is consistent with the *unrolled iterative estimation view.*

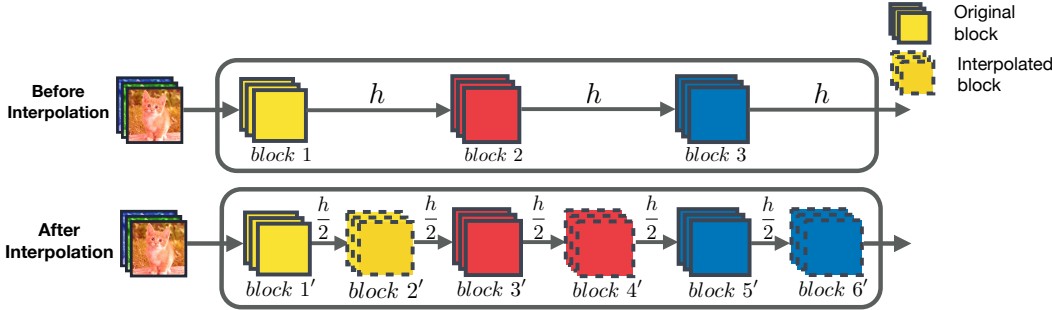

Figure 5: **An illustration of the interpolation operation for one stage.** We insert one residual block right after each existing block in the stage. The model parameters, including convolutional weights and batch normalization parameters, are copied from the adjacent old block to interpolated blocks. After that, the explicit step size $h$ is halved. For example, before interpolation, this stage has three residual blocks, numbered 1 to 3. After interpolation, block 1, 2 and 3 become block 1', 3' and 5' respectively. Three new blocks are inserted: block 2', 4' and 6', whose parameters are copied from its previous block respectively.

|  | # Residual Blocks | Explicit Step Size $h$ | # Training Steps |
|---|---|---|---|
| **Cycle 1** | 2-2-2 | 1 | $N_1$ |
| **Cycle 2** | 4-4-4 | 0.5 | $N_2$ |
| **Cycle 3** | 8-8-8 | 0.25 | $N_3$ |

Table 1: **An illustration of the multi-level method with 3 cycles.** The ResNet model has 3 stages; *# Residual Blocks* column represents the number of blocks in each stage. In cycle 1, the training starts with a 2-2-2 model using $h = 1$. After $N_1$ training steps, the first interpolation happens: the model becomes 4-4-4, and the step size is halved to 0.5. Similarly, $N_2$ training steps later, the second interpolation doubles the number of blocks to 8-8-8 and halves $h$ to 0.25. Cycle 3 lasts for $N_3$ training steps.

## 4 EFFICIENT TRAINING WITH MULTI-LEVEL METHOD

Given the connection between ResNets and ODEs, existing theories and numerical techniques for ODEs can be applied to ResNets. In numerical analysis, multi-grid methods (Hackbusch, 2013) are algorithms for solving differential equations using a hierarchy of discretizations with varying step sizes. Inspired by multi-grid methods, we propose the *multi-level training method*.

### 4.1 MULTI-LEVEL TRAINING

The idea of the multi-level method is, during training, we start with a shallow network using a large explicit step size $h$. After a few training steps, we switch to a deeper network, by doubling the number of residual blocks and halving the step size to $h/2$. This operation is called *interpolation*, which applies to all the stages at the same time. Fig. 5 illustrates the interpolation operation for one stage. The interpolation operation inserts a new residual block right after every existing block, and copies the convolutional weights and batch normalization parameters from the adjacent old block to the new block.

In the multi-level training process, interpolations happen several times, thus dividing the training steps into *cycles*. Table 1 gives an example to illustrate this process. According to our dynamical systems view, by interpolating the residual blocks and halving the step size, we solve exactly the same differential equation. Therefore, the interpolation operation at the beginning of a cycle gives a good initialization of the parameters.

Each cycle itself can be regarded as a training process, thus we need to reset the learning rate to a large value at the beginning of each training cycle and anneal the learning rate during that cycle. Here we adopt the cosine annealing learning rate schedule (Loshchilov & Hutter, 2017). Within

| # Interpolations | 0 | 1 | 2 | 3 | 4 | 5 | $\cdots$ |
|---|---|---|---|---|---|---|---|
| **Theoretical Time Saved** | 0% | 25% | 42% | 53% | 61% | 67% | $\cdots$ |

Table 2: **Number of interpolations vs theoretical time saved, relative to the full model.** Theoretically, time saved is monotonically increasing as the number of interpolation increases, but the marginal benefit is diminishing.

each cycle, the learning rate is

$$\eta = \eta_{\min} + \frac{1}{2}(\eta_{\max} - \eta_{\min})(1 + \cos(\frac{T_{\text{cur}}}{T}\pi)), \tag{9}$$

where $\eta_{\min}$ and $\eta_{\max}$ represent the minimum and maximum learning rate respectively, $T_{\text{cur}}$ accounts for how many training steps have been performed in the current cycle, and $T$ denotes the total number of training steps in this cycle. The learning rate starts from $\eta_{\max}$ at the beginning of each cycle and decreases to $\eta_{\min}$ at the end of the cycle.

## 4.2 TRAINING TIME

Since the number of residual blocks in cycle $i$ is half of that in cycle $i+1$, theoretically, the running time in cycle $i$ should also be half of that in cycle $i+1$. Take a multi-level method with two cycles as an example, it trains a shallow model (2-2-2 blocks) for $N$ steps and switches to a deep model (4-4-4 blocks) for another $N$ steps. Compared with the deep model trained for $2N$ steps, the multi-level method reduces training time by $1/4$.

More generally, if one uses the multi-level method with $k$ interpolations equally dividing the training steps, theoretically it saves $1 - \frac{2^{k+1}-1}{2^k(k+1)}$ of training time, compared to the full model (model in the last cycle) trained for the same number of total steps. Table 2 shows the theoretical time saved. Time saved is monotonically increasing as the number of interpolation increases, but the marginal time saved is diminishing. Furthermore, when the number of interpolations is large, each cycle might not have enough training steps. Therefore, there is a trade-off between efficiency and accuracy.

## 5 EXPERIMENTS

The empirical results on the dynamical systems view are presented in Sec. 3.2 and 3.3. In this section, we evaluate the efficacy and efficiency of the proposed multi-level method on two state-of-the art deep learning architectures for image classification: ResNet and Wide ResNet, across three standard benchmarks.

| Model | # Blocks | CIFAR-10 | | CIFAR-100 | | STL-10 | |
|---|---|---|---|---|---|---|---|
| | | Error | Time | Error | Time | Error | Time |
| ResNet-14 | 2-2-2 | 9.75% | 38m | 33.34% | 38m | 27.78% | 33m |
| ResNet-50 | 8-8-8 | 7.58% | 114m | 28.64% | 115m | 25.95% | 114m |
| ResNet-50-i (**Ours**) | 2-2-2 to 8-8-8 | 7.10% | 67m | 28.71% | 68m | 25.98% | 68m |
| ResNet-32 | 5-5-5 | 7.74% | 76m | 29.96% | 74m | 26.02% | 71m |
| ResNet-122 | 20-20-20 | 6.47% | 266m | 26.74% | 266m | 25.16% | 266m |
| ResNet-122-i (**Ours**) | 5-5-5 to 20-20-20 | 6.56% | 154m | 26.81% | 154m | 24.36% | 162m |

Table 3: **Main multi-level method results for ResNets with different depths.** The model name with $i$ corresponds to the multi-level method. Our multi-level training method achieves superior or on-par accuracy with the last cycle model while saving about 40% of training time. The unit of training time is a minute.

## 5.1 DATASETS AND NETWORKS

**Datasets** Three widely used datasets are used for evaluation: CIFAR-10, CIFAR-100 (Krizhevsky & Hinton, 2009), and STL10 (Coates et al., 2011). Details on these datasets and data augmentation methods can be found in Appendix A.

**Networks** We use ResNets (He et al., 2016) and Wide ResNets (Zagoruyko & Komodakis, 2016) for all the datasets. All the networks have three stages, with the number of filters equal to 16-32-64 for ResNets, and 32-64-128 for Wide ResNets.

## 5.2 EXPERIMENTAL SETTINGS

Based on the analysis in Table 2, we use two interpolations, that is *three cycles*, for the multi-level method in order to optimize the trade-off between efficiency and accuracy.

For each experiment, we run our multi-level model with *three cycles*. For comparison, two other models are trained for the same number of steps: a model with the same architecture as the *first cycle* and a model with the same architecture as the *last cycle*.

We call them *first cycle model* and *last cycle model* respectively. We also use the cyclic learning rate schedule (Loshchilov & Hutter, 2017) for the *first cycle model* and *last cycle model* for fair comparison.

All the models are trained for 160 epochs. For our multi-level method, the models are interpolated at the 60th and 110th epochs. For baseline models, the learning rate cycle also restarts at epoch 60 and 110. The maximum and minimum learning rates $\eta_{min}$ and $\eta_{max}$ are set 0.001 and 0.5 respectively. For CIFAR-10 and CIFAR-100 experiments, the mini-batch size is 100. For STL-10 experiments, the mini-batch size is 32. We use a weight decay of $2 \times 10^{-4}$, and momentum of 0.9. All the experiments are evaluated on machines with a single Nvidia GeForce GTX 1080 GPU. The networks are implemented using TensorFlow library (Abadi et al., 2016).

## 5.3 MAIN RESULTS AND ANALYSIS

We present the main results and analysis in this section. More experimental results can be found in Appendix D. The theoretical time saved for two interpolations is 42%, which is consistent with the experiment results.

The main results are shown in Table 3 and 4, for ResNets and Wide ResNets respectively. We report the test error rate and training time. Compared with the first cycle model, our multi-level method achieves much lower test error. Compared with the last cycle model, the test error is competitive or slightly lower, but the training time reduction is over 40%. This result applies to both ResNets and WResNets across three datasets. The interpolation of (Wide) ResNet-50-i from 2-2-2 to 8-8-8 and (Wide) ResNet-122-i from 5-5-5 to 20-20-20 show that our multi-level training method is effective for different network depths.

The train and test curves with both ResNets and Wide ResNets are shown in Fig. 6. Although both training and test accuracy temporarily drops at the start of each cycle, the performance eventually surpasses the previous cycles. ResNets and Wide ResNets have similar train and test curves, indicating that our multi-level training method is effective for different network widths.

## 6 CONCLUSION

In this work, we study ResNets from the dynamical systems view and explain the lesion studies from this view through both theoretical and empirical analyses. Based on these analyses, we develop a simple yet effective multi-level method for accelerating the training of ResNets. The proposed multi-level training method is evaluated on two state-of-the-art residual network architectures across three widely used classification benchmarks, reducing training time by more than 40% with similar accuracy. For future work, we would like to explore the dynamical systems view on other ResNets variants such as DenseNets.

| Model | # Blocks | CIFAR-10 | | CIFAR-100 | | STL-10 | |
|---|---|---|---|---|---|---|---|
| | | Error | Time | Error | Time | Error | Time |
| WResNet-14 | 2-2-2 | 7.38% | 51m | 27.92% | 51m | 24.58% | 63m |
| WResNet-50 | 8-8-8 | 5.87% | 174m | 24.49% | 173m | 23.82% | 222m |
| WResNet-50-i (**Ours**) | 2-2-2 to 8-8-8 | 5.95% | 101m | 24.92% | 101m | 22.82% | 131m |
| WResNet-32 | 5-5-5 | 6.29% | 111m | 25.32% | 111m | 23.51% | 136m |
| WResNet-122 | 20-20-20 | 5.38% | 406m | 23.11% | 406m | 22.00% | 516m |
| WResNet-122-i (**Ours**) | 5-5-5 to 20-20-20 | 5.46% | 239m | 23.04% | 237m | 22.65% | 307m |

Table 4: **Main multi-level method results for Wide ResNets (WResNets) with different depths.** The model name with *i* corresponds to the multi-level method. Our multi-level training method achieves superior or on-par accuracy with the last cycle model while saving about 40% of training time. The unit of training time is a minute.

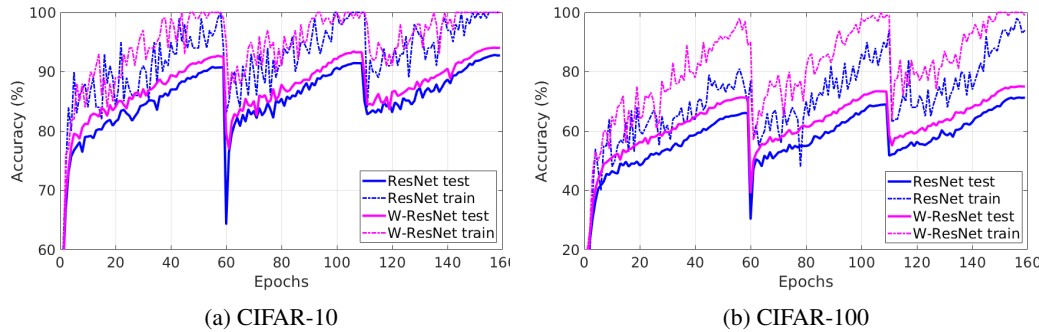

(a) CIFAR-10         (b) CIFAR-100

Figure 6: **Train and test curves using our multi-level method with ResNet-50-i and WResNet-50-i on CIFAR-10/100.** The models are interpolated at epoch 60 and 110, dividing the training steps to three cycles. Although both training and test accuracy temporarily drops at the start of each cycle, the performance eventually surpasses the previous cycles.

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

## A    DATASETS AND MODEL DETAILS

**CIFAR-10 and CIFAR-100**  The CIFAR-10 dataset (Krizhevsky & Hinton, 2009) consists of 50,000 training images and 10,000 testing images in 10 classes with $32 \times 32$ image resolution. The CIFAR-100 dataset has the same number of training and testing images as CIFAR-10, but has 100 classes. We use the common data augmentation techniques including padding four zeros around the image, random cropping, random horizontal flipping and image standardization (Huang et al., 2016).

**STL-10**  The STL-10 dataset (Coates et al., 2011) is an image classification dataset with 10 classes with image resolutions of $96 \times 96$. It contains 5,000 training images and 8,000 testing images. Compared with CIFAR-10/100, each class has fewer training samples but higher image resolution. We use the same data augmentation as the CIFAR-10/100 except for padding 12 zeros around the images.

**Number of parameters**  We list the number of parameters for each model in Table 5.

| Model | # Blocks | CIFAR-10 | CIFAR-100 | STL-10 |
|---|---|---|---|---|
| ResNet-14 | 2-2-2 | 172,506 | 178,356 | 172,506 |
| ResNet-50 | 8-8-8 | 755,802 | 761,652 | 755,802 |
| ResNet-32 | 5-5-5 | 464,154 | 470,004 | 464,154 |
| ResNet-122 | 20-20-20 | 19,22,394 | 1,928,244 | 19,22,394 |
| WResNet-14 | 2-2-2 | 685,994 | 697,604 | 685,994 |
| WResNet-50 | 8-8-8 | 3,013,802 | 3,025,412 | 3,013,802 |
| WResNet-32 | 5-5-5 | 1,849,898 | 1,861,508 | 1,849,898 |
| ResNet-122 | 20-20-20 | 7,669,418 | 7,681,028 | 7,669,418 |

Table 5: **The number of parameters for each network model.**

## B    IMPLEMENTATION DETAILS

**Interpolation** Figure 5 gives a conceptual illustration of the interpolation operation. When implementing this operation, the first block in each stage requires special treatment. The first block changes the size of the feature map and the number of channels, thus the shapes of convolution and batch normalization parameters are different from those in subsequent blocks. As a result, we cannot simply copy the parameters from block 1' to block 2'. Instead, the parameters of block 2' are copied from block 3'.

## C    FURTHER ANALYSIS OF TIME STEP SIZE

Instead of using $h$ as an implicit step size in Sec. 3.2, we can also explicitly express it in the model. That is, a residual block represents the function $F$ instead of $G$, and $h$ becomes a hyperparameter of the model. In other words, in each residual block, the output of the residual module is multiplied by $h$ before being added to the identity mapping. Now $h$ is the **explicit step size** that we can control. This enables us to verify the claim that *the depth of the network does not affect the underlying differential equation.*

**Empirical analysis** We run the following experiments on CIFAR-10 and CIFAR-100:

- ResNet-32: 15 blocks with $h = 1$;
- ResNet-62: 30 blocks with $h = 0.5$;
- ResNet-122: 60 blocks with $h = 0.25$;

According to our theory, those three models are different discretizations of the same differential equation in Eq. (2). As a result, the function values $F(t)$ should be roughly the same across the time interval $[0, T]$, which is discretized to 15, 30 and 60 time steps respectively. In Figure 7, we plot the $L^2$-norm of the residual blocks $F(\mathbf{Y}_j)$ and scale the block number to the corresponding conceptual time points. It can be seen from the figure that the three curves follow the same trend, which represents the underlying function $F(t)$.

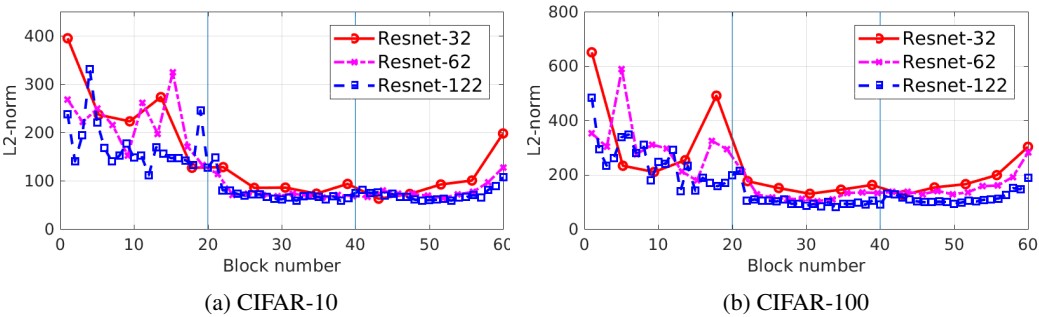

(a) CIFAR-10                                (b) CIFAR-100

Figure 7: **Comparison of $\|F(\mathbf{Y}_j)\|$ among three models**: (1) ResNet-32 with $h = 1$, (2) ResNet-62 with $h = 0.5$, (3) ResNet-122 with $h = 0.25$. If the dynamical systems view is correct, the three curves should approximately follow the same trend.

## D    MORE EXPERIMENTAL RESULTS

**Train and test curves on STL-10** We show the train and test curves using our multi-level training method on ResNet-50-i and Wide ResNet-50-i on STL10 in Fig. 8.

**Effect of learning rate** To study the effect of $\eta_{\max}$ and $\eta_{\min}$ on the cyclic learning rate in Eq. (9), we plot their effect on test accuracy in Fig. 9. We empirically find that $\eta_{\max} = 0.5$ and $\eta_{\min} = 0.001$ achieve the best accuracy. With the increase of maximum learning rate $\eta_{\max}$, the test accuracy increases first and then decreases. A potential explanation is that with a small learning rate, the

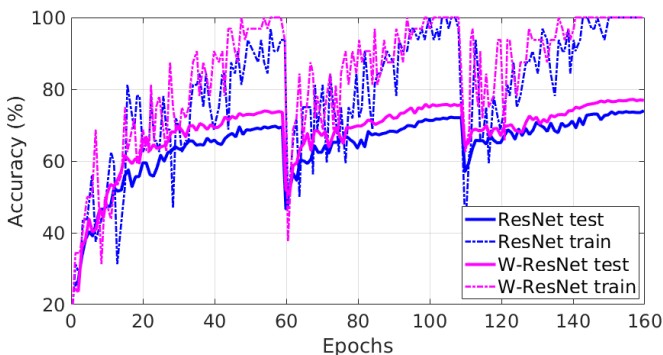

Figure 8: **Train and test curves using our multi-level method with ResNet-50-i and WResNet-50-i on STL10.** Although both training and testing accuracy temporarily drops at the start of each cycle, the performance eventually surpasses the previous cycles.

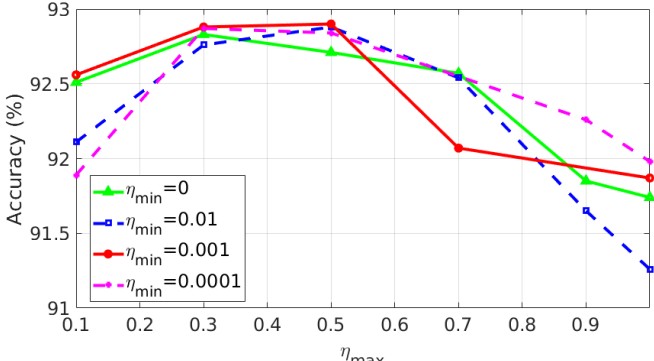

Figure 9: **Effect of $\eta_{\max}$ and $\eta_{\min}$ in cyclic learning rate.** In most cases, as $\eta_{\max}$ increases, the testing accuracy increases first and then decreases.

system learns slowly, while with a high learning rate, the system contains too much kinetic energy and is unable to reach the deeper and narrower parts of the loss function.

**Effect of resetting the learning rate** To study the effect of resetting the learning rate at the beginning of each cycle, we also run the multi-level training method without resetting the learning rate, that is to use Eq. 9 throughout all cycles. The experiment setting is the same as described in Section 5.2. We train Resnet-50-i on CIFAR-10, CIFAR-100 and STL-10. Each setting is repeated 10 times to obtain the confidence intervals. Fig. 10 shows the results: resetting the learning rate at the beginning of each cycle gives better validation accuracy.

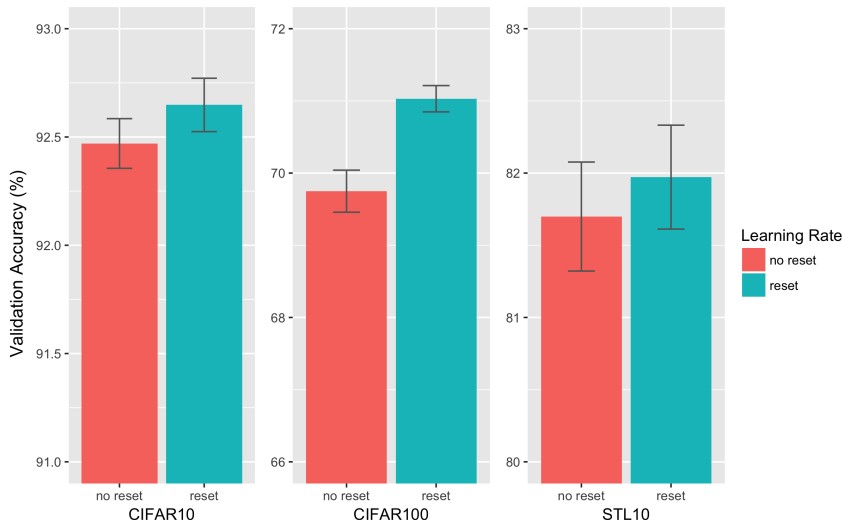

Figure 10: **Effect of resetting the learning rate.** Resetting the learning rate at the beginning of each cycle gives better validation accuracy.

**Comparison with shallow and deep ResNets** Fig. 11 shows that training accuracy curves for a shallow ResNet (same depth as the starting model) and a deep ResNet (same depth as the final model).

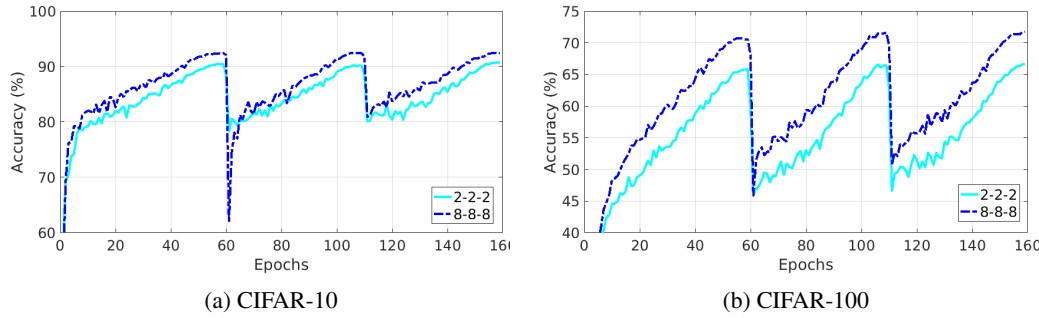

(a) CIFAR-10               (b) CIFAR-100

Figure 11: **Shallow and deep ResNets.** Training accuracy curves for a shallow ResNet (same depth as the starting model) and a deep ResNet (same depth as the final model).

