# OpenReview forum: "Multi-level Residual Networks from Dynamical Systems View"
_ICLR.cc/2018/Conference — Accept (Poster)_

### Official Review · AnonReviewer1 · 2017-11-26

**Rating:** 7
**Confidence:** 4

**Review:**

This paper interprets deep residual network as a dynamic system, and proposes a novel training algorithm to train it in a constructive way. On three image classification datasets, the proposed algorithm speeds up the training process without sacrificing accuracy. The paper is interesting and easy to follow.

I have several comments:
1.	It would be interesting to see a comparison with Stochastic Depth, which is also able to speed up the training process, and gives better generalization performance. Moreover, is it possible to combine the proposed method with Stochastic Depth to obtain further improved efficiency?
2.	The mollifying networks [1] is related to the proposed method as it also starts with shorter networks, and ends with deeper models. It would be interesting to see a comparison or discussion.
[1] C Gulcehre, Mollifying Networks, 2016
3.	Could you show the curves (on Figure 6 or another plot) for training a short ResNet (same depth as your starting model) and a deep ResNet (same depth as your final model) without using your approach?

---

> ### Author Response · Authors · 2018-01-05
> **Response to AnonReviewer1**
>
> We would like to thank the reviewer for the detailed comments and suggestions for the manuscript.
>
> (1) Lu et al. [1] introduced stochastic dynamic system perspective and interpreted Stochastic Depth method as an approximation to a stochastic dynamic system. Combining multi-level method with stochastic dynamic system view is one of our future research directions.
>
> (2) We thank the reviewer for pointing out the mollifying network paper. We have added it in the related work section. The mollifying network starts with a linearized network with a smoothed objective function, and evolves to a non-linear network and the original objective function. Both mollifying network and our proposed method go from simple networks to complex ones. But mollifying network solves a smoothed problem, while our method solves the same underlying differential equations, but at different levels of approximation. Also, the purpose of mollifying network is to make the optimization easier, while ours is to speed up training.
>
> (3) Short and deep ResNet curves: Please see Fig.11 in appendix D in the updated manuscript.
>
> [1] Lu, Yiping, et al. "Beyond Finite Layer Neural Networks: Bridging Deep Architectures and Numerical Differential Equations." arXiv preprint arXiv:1710.10121 (2017).

---

### Official Review · AnonReviewer3 · 2017-11-27
**Pleasant to read, well executed paper.**

**Rating:** 7
**Confidence:** 3

**Review:**

I enjoyed reading the paper. This is a very well written paper, the authors propose a method for speeding up the training time of Residual Networks based on the dynamical system view interpretation of ResNets. In general I have a positive opinion about the paper, however, I’d like to ask for some clarifications.

I’m not fully convinced by the interpretation of Eq. 5: “… d is inversely proportional to the norm of the residual modules G(Yj)”. Since F(Yj) is not a constant, I think that d is inversely proportional to ||G(Yj)||/||F(Yj)||, however, in the interpretation the dependence on ||F(Yj)|| is ignored. Could the authors comment on that?

Section 4. 1 “ Each cycle itself can be regarded as a training process, thus we need to reset the learning rate value at the beginning of each training cycle and anneal the learning rate during that cycle.” Is there any empirical evidence for this? What would happen if the learning rate is not reset at the beginning of each cycle?

Questions with respect to dynamical systems point of view: Eq. 4 assumes small value of h. However, for ResNet there is no guarantee that the h would be small (e. g. in Appendix C the values between 0.25 and 1 are used). Would the authors be willing to comment on the importance of the value of h? In figure 1, pooling (strided convolutions) are not depicted between network stages. I have one question w.r.t. feature maps dimensionality changes inside a CNN: how does pooling (or strided convolution) fit into dynamical systems view?

Table 3 and 4. I assume that the training time unit is a minute, I couldn’t find this information in the paper. Is the batch size the same for all models (100 for CIFAR and 32 for STL-10)? I understand that the models with different #Blocks have different capacity, for clarity, would it be possible to add # of parameters to each model? For multilevel method, would it be possible to show intermediate results in Table 3 and 4, e. g. at the end of cycle 1 and 2? I see these results in Figure 6, however, the plots are condensed and it is difficult to see the exact number at the end of each cycle.

The citation (E, 2017) seems to be wrong, could the authors check it?

---

> ### Author Response · Authors · 2018-01-05
> **Response to AnonReviewer3**
>
> We would like to thank the reviewer for the detailed comments and suggestions for the manuscript.
>
> (1) According to our theoretical interpretation, F represents the underlying ODE and does not depend on d. In Appendix C, we also empirically show that the depth of the network does not affect the underlying differential equation.
>
> (2) Resetting learning rate: We ran experiments comparing resetting and not resetting the learning rate at the beginning of each cycle. The results are shown in Appendix D, Figure 10. Resetting the learning rate at the beginning of each cycle gives better validation accuracy in the updated manuscript.
>
> (3) The value of h: In this paper, we formulate ResNet as a forward Euler discretization of an ODE. For forward Euler, h times the norm of convolution kernels should to be small to ensure stability. In practice, h is absorbed by the convolution kernels and stability is achieved by regularizing the convolution kernels.
>
> (4) Moving between different blocks in the network is equivalent to changing the resolution when solving a time dependent differential equation. Algorithms commonly use high resolution at early times and coarsen the image for later times, similar to different units of the ResNet.
>
> (5) Number of parameters: Please see Table 5 in appendix A in the updated manuscript.
>
> (6) E is the last name of Weinan E (https://web.math.princeton.edu/~weinan/).

---

### Official Review · AnonReviewer2 · 2017-11-29
**Review of "Multi-level Residual Networks from Dynamical Systems View"**

**Rating:** 7
**Confidence:** 4

**Review:**



This paper proposes a new method to train residual networks in which one starts by training shallow ResNets, doubling the depth and warm starting from the previous smaller model in a certain way, and iterating.  The authors relate this idea to a recent dynamical systems view of ResNets in which residual blocks are viewed as taking steps in an Euler discretization of a certain differential equation.  This interpretation plays a role in the proposed training method by informing how the “step sizes” in the Euler discretization should change when doubling the depth of the network.  The punchline of the paper is that the authors are able to achieve similar performance as “full ResNet training” but with significantly reduced training time.

Overall, the proposed method is novel — even though this idea of going from shallow to deep is natural for residual networks, tying the idea to the dynamical systems perspective is elegant.  Moreover the paper is clearly written.  Experimental results are decent — there are clear speedups to be had based on the authors' experiments.  However it is unclear if these gains in training speed are significant enough for people to flock to using this (more complicated) method of training.

I only have a few small questions/comments:
* A more naive way to do multi-level training would be to again iteratively double the depth, but perhaps not halve the step size.  This might be a good baseline to compare against to demonstrate the value of the dynamical systems viewpoint.
* One thing I’m unclear on is how convergence was assessed… my understanding is that the training proceeds for a fixed number of epochs (?) - but shouldn’t this also depend on the depth in some way?
* Would the speedups be more dramatic for a larger dataset like Imagenet?
* Finally, not being very familiar with multigrid methods from the numerical methods literature — I would have liked to hear about whether there are deeper connections to these methods.

---

> ### Author Response · Authors · 2018-01-05
> **Response to AnonReviewer2**
>
> We would like to thank the reviewer for the detailed comments and suggestions for the manuscript.
>
> (1) According to the dynamical systems view, by halving the step size, the underlying differential system is the same before and after interpolation. Section 3.2 and Appendix C validate this interpretation empirically. In practice, the step size h will be absorbed by the convolution kernels over the course of normal backpropagation if it is not properly halved during multi-level training.
>
> (2) Yes, we used a fixed number of epochs for different models. You are right that technically the training steps should be dependent on the depth in some way. However, in the literature, researchers commonly use fixed number of epochs to compare models with varying depths or size. For example in [1], training terminates at 64k iterations on CIFAR-10 for all models with number of layers ranging from 20 to 1202.
>
> (3) We are currently working on the experiments of ImageNet, and we are trying our best to include the results in the final version.
>
> (4) Our methods are closely linked to grid continuation techniques. These techniques are commonly used in optimal control problems in the context of fluid flow and path planning. The basic idea is to use the continuous underlying structure (the pde or ode) in order to gradually discretize the problem on a increasingly finer mesh. See [2] for more details.
>
> [1] He, Kaiming, et al. "Deep residual learning for image recognition." Proceedings of the IEEE conference on computer vision and pattern recognition. 2016.
>
> [2] E. Allgower and K. Georg, Numerical continuation methods, Springer Verlag, 1990.

---

> > ### Public Comment · (anonymous) · 2018-01-26
> > **ImageNet results ?**
> >
> > > We are currently working on the experiments of ImageNet
> >
> > Any update on this front ?
> > Improved ImageNet training time would significantly increase the impact of this paper.

---

### Author Response · Authors · 2018-01-05
**Revision: model details and additional experimental results**

Dear reviewers,

Thanks for your comments and suggestions. We have upload a revision and added model details in Appendix A and more experimental results in Appendix D.

---

### Decision · Program_Chairs · 2018-01-29
**ICLR 2018 Conference Acceptance Decision**

**Decision:**

Accept (Poster)

**Comment:**

this submission proposes a learning algorithm for resnets based on their interpreration of them as a discrete approximation to a continuous-time dynamical system.  all the reviewers have found the submission to be clearly written, well motivated and have proposed an interesting and effective learning algorithm for resnets.